# Catalytic Hydrogen Production from Methane: A Review on Recent Progress and Prospect

**Luning Chen [1],[†], Zhiyuan Qi [1],[†], Shuchen Zhang [1] , Ji Su [1],[2],* and Gabor A. Somorjai [1],[3],***

[1]   Materials Sciences Division, Lawrence Berkeley National Laboratory, Berkeley, CA 94720, USA; luningchen@lbl.gov (L.C.); zhiyuan7@lbl.gov (Z.Q.); shuchenzhang@lbl.gov (S.Z.)
[2]   Molecular Foundry, Material Science Division, Lawrence Berkeley National Laboratory, Berkeley, CA 94720, USA
[3]   Department of Chemistry, University of California-Berkeley, Berkeley, CA 94720, USA
*    Correspondence: jisu@lbl.gov (J.S.); somorjai@berkeley.edu (G.A.S.); Tel.: +1-(510)-642-4053 (G.A.S.)
†    These authors contributed equally to this work.

**Abstract:** Natural gas (Methane) is currently the primary source of catalytic hydrogen production, accounting for three quarters of the annual global dedicated hydrogen production (about 70 M tons). Steam–methane reforming (SMR) is the currently used industrial process for hydrogen production. However, the SMR process suffers with insufficient catalytic activity, low long-term stability, and excessive energy input, mostly due to the handling of large amount of $CO_2$ coproduced. With the demand for anticipated hydrogen production to reach 122.5 M tons in 2024, novel and upgraded catalytic processes are desired for more effective utilization of precious natural resources. In this review, we summarized the major descriptors of catalyst and reaction engineering of the SMR process and compared the SMR process with its derivative technologies, such as dry reforming with $CO_2$ (DRM), partial oxidation with $O_2$, autothermal reforming with $H_2O$ and $O_2$. Finally, we discussed the new progresses of methane conversion: direct decomposition to hydrogen and solid carbon and selective oxidation in mild conditions to hydrogen containing liquid organics (i.e., methanol, formic acid, and acetic acid), which serve as alternative hydrogen carriers. We hope this review will help to achieve a whole picture of catalytic hydrogen production from methane.

**Keywords:** hydrogen economic; methane conversion; heterogeneous catalysts

## 1. Introduction

Methane ($CH_4$) is an important chemical feedstock for hydrogen production. In the United States, more than 95% of the hydrogen is produced by steam–methane reforming (SMR) with an annual hydrogen production of 10 million metric tons. SMR is a mature industrial process to harvest $H_2$ from a methane source (e.g., natural gas) under high-temperature steam (700–1000 °C) with a pressure range of 3–25 bar. Besides $H_2$, carbon monoxide (CO), and a relatively small amount of carbon dioxide ($CO_2$) were also obtained from SMR, which was therefore developed to manufacture syngas ($H_2$ and CO) traditionally. When coupled with a water–gas shift (WGS) reaction, the co-product CO from SMR could further react with steam to result in one extra $H_2$ and $CO_2$ as the final products [1–3]. In general, SMR could be considered as a catalytic process that accelerates the decomposition of methane to $H_2$ gas and carbon species. The C species were then further reacted with oxygen via a gasification reaction to generate $CO/CO_2$ and recover the active sites.

Although SMR is a mature technology, it still suffers from several disadvantages caused by the reactant properties and reaction thermodynamics: High energy consumption, high production cost, harsh reaction conditions, low reaction efficiency, and low process stability [4,5]. Specifically,

the disadvantages are mainly reflected in three aspects: Firstly, methane is very stable, and thus hard to activate. Due to the highly endothermic nature ($\Delta Ho$ = 206 kJ/mol), SMR requires extra energy and designed instruments to proceed the reaction at high temperature and pressure, which also introduces mass transfer and heat transfer issues. Secondly, the design of catalysts still needs improvement. On the one hand, noble metal catalysts (e.g., Ru, Rh) displayed high SMR activities with good stability. However, the cost and availability of noble metals limit their application [6–8]. On the other hand, the commercial catalyst $Ni/Al_2O_3$ suffers serious deactivation, due to the easy sintering and coke formation of Ni catalysts. Thirdly, SMR coproduces $CO_2$, especially when coupled with a WGS for $H_2$ production (9–14 kg $CO_2$/kg$H_2$). The $CO_2$ has a strong greenhouse effect and its handling further increases the cost of the SMR process [9].

To develop a highly efficient, low cost and stable methane reforming process, the research effects focus on three directions. First of all, alternative and upgraded techniques for methane conversion were developed including dry reforming with carbon dioxide (DRM) [10–12], partial oxidation with $O_2$ [13,14], autothermal reforming with $H_2O$ and $O_2$ [15], low temperature SMR [16], combined steam and dry reforming of methane (CSDRM) [17], a chemical looping SMR process (CL-SMR) [18], a sorption-enhanced SMR (SE-SMR) [19–21], as well as a combined CL-SE-SMR process [22]. Each process generates a syngas mixture with an alternating $H_2$:CO ratio, due to different reaction conditions applied such as temperature and pressure. Furthermore, a series of catalysts with higher activity and stability were designed and fabricated. The most investigated catalysts in this field are mostly from Group VIII, including monometallic, bimetallic noble and non-noble metal catalysts [1]. The applied metals [23], size effect [24–27], promoter effect [1,28], support effect [29–31], as well as coordination state [32] and acidic/basic properties [33] of these catalysts were studied in-depth. The catalysts applied and above-mentioned variables are essential for the control of the reaction routes and the production distributions. Finally, reaction conditions and various engineering factors of these methane reforming processes were further optimized, for example, temperature, pressure, feeding rate, feedstock ratio, reactor types etc.

## 2. Descriptors of Structure–Activity Relationship of the Catalyst

The descriptors of the structure–activity relationship of the catalyst such as metal type, metal coordinate state, size effect, support effect, and promoters, are an important interface to bridge the gap between deep learning and observed catalytic performance. Previous research work reveals that both SMR and DRM are structure sensitive [34–36]. Clearly, atoms with different chemical environments in the catalytic metal nanoparticles have distinct activities [24]. Therefore, mapping this structure sensitivity and understanding the associated mechanism provides opportunities to design more efficient and stable catalysts.

### 2.1. Metal Type Effect

One major type of steam reforming catalysts is noble metal-based. Unlike commercial Ni catalysts that have serious coking problem caused by the formation, diffusion, and dissolution of carbon in Ni metals, noble metals yield much less coking due to the difficulties of dissolving carbon in them [37]. Noble metals such as Ru, Rh, Pd, Ir, and Pt were examined for their reforming performance, among which Ru and Rh displayed high reforming activities and low carbon formation rates [38,39]. Jones et al. [25]. proposed the SMR activity order as Ru > Rh > Ir > Pt, indicating Pt catalyst as the least active among the other metals. In contrast, the study of Wei and Iglesia [26] showed that C–H bond activation by Pt is more efficient than Ir, Rh, and Ru (Figure 1). Therefore, due to the disagreement from different research groups involved in this field, it is still not possible to give a specific order of catalytic activity and selectivity towards hydrogen production via SMR.

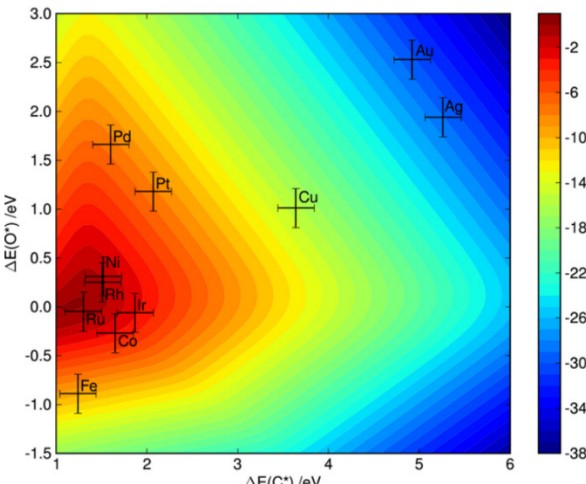

**Figure 1.** Two-dimensional volcano-curve of the turn over frequency (log10) as a function of O and C adsorption energy. T = 773 K, P = 1 bar; 10% conversion. The error bars include an estimated 0.2 eV uncertainties in the adsorption energies [26].

Although all these noble metal catalysts show high catalytic activity and less coke formation, the high cost hinders their practical applications. Group VIII non-noble metals are also active for SMR [40–42]. However, iron will be rapidly oxidized under reaction conditions while cobalt cannot withstand the partial pressures of steam. Therefore, nickel-based catalysts are the most studied and commonly used SMR catalysts at industrial scale mostly owing to their low cost.

### 2.2. Ni Size Effect

In this section, we will mainly summarize the size effect of Ni catalysts [43], considering the commercialization and extensive research work on Ni (60% of publications in the field of SMR are about Nickel catalysts [1]). The optimal Ni particle size for both SMR and DMR at 500 and 600 °C under 5 bar, was found to be approximately 2−3 nm, whereas carbon whisker formation was found to significantly occur on ~4.5 nm Ni particles during SMR and increased with increasing particle size during DRM [24].

Furthermore, the performances of single site $Ni_1$/MgO catalysts for DRM reaction was systemically studied by combining theoretical modeling (Density Functional Theory (DFT) and Kinetic Monte Carlo Method (KMC) simulations) and experimental results. The DFT calculations show that synergistic effect between Ni single atom and MgO support in $Ni_1$/MgO is not strong due to the weak binding of reaction intermediates and the limited numbers of neighboring active sites. When slightly increasing the Ni size, the single site $Ni_4$/MgO catalyst is able to provide stronger bindings than $Ni_1$/MgO. It also offers enough active but isolated Ni sites working cooperatively for the activation of both $CH_4$ and $CO_2$, which produces CO, $H_2$ and $H_2O$ while completely eliminating carbon deposition. The experimental observation on a 5% Ni/MgO catalyst including a Ni cluster of 3–4 Ni atoms agrees well with the calculation predictions (Figure 2) [27].

### 2.3. Promoter Effect

The researchers showed that the addition of second metal to a Ni-based catalyst can improve its selectivity, durability, and activity, thus limiting the typical problems of SMR including coke formation, active oxidation, sintering, and segregation [44–47]. For example, combining of the highly reactive Ni species for $CH_4$ activation and Fe species for water splitting, together with the resulting Ni–Fe alloy, achieved a high $CH_4$ conversion up to 97.5% and CO selectivity up to 92.9% at 900 °C with productivity of CO and $H_2$ of 9.6 and 29.0 mol kg catalyst$^{-1}$, respectively (on equimolar Ni–Fe catalyst) [21].

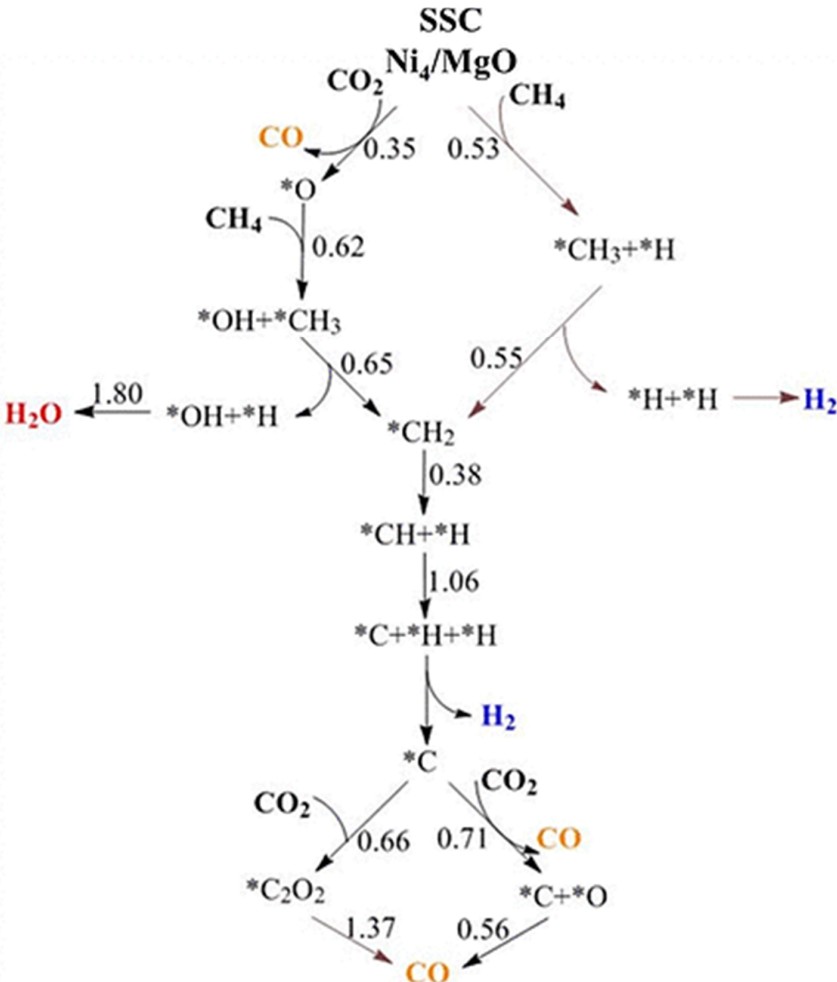

**Figure 2.** Schematics of KMC-identified reaction network for the dry reforming with $CO_2$ (DRM) on the confined single-site $Ni_4/MgO(100)$ catalyst. The corresponding Ea for each step is also included (in eV) [27].

### 2.4. Support Effect

A large number of studies were focused on the improvement of Ni performance via analyzing the effect of different supports [48–51]. Nieva et al. stated that the catalytic activity of Ni-based catalysts mainly depends on the support, which plays an important role in the catalytic process. In fact, the supports influence the metal dispersion, affect sintering resistance, and sometimes directly participate in reactions by facilitating the adsorption of reactants. For SMR at 600 °C, the activity order of Ni-based catalysts with different supports was observed as: $Ni/MgAl_2O_4 > Ni/ZnAl_2O_4 > Ni/Al_2O_3 > Ni/SiO_2$. Specifically, $Ni/SiO_2$ undergoes a rapid deactivation probably due to the surface oxidation and carbon deposition, whereas $Ni/ZnAl_2O_4$ shows the lowest degree of carbon deposition and the highest resistance to sintering [29].

In addition, it was observed that supports such as $Al_2O_3$ and $SiO_2$ allow a gradual oxidization of Ni catalysts during SMR at 500 °C under ambient pressure, whereas the utilization of $ZrO_2$ stabilized Ni particles and allowed for higher methane conversion than aforementioned supports. It is mainly because the $ZrO_2$ support allows a water accumulation, favoring the formation of hydroxyl groups which promotes SMR [30]. $CeO_2$ has also been widely studied as both a support and promoter of Ni-based catalysts because of its high thermal stability, mechanical resistance, and high oxygen storage capacity. Dan et al. studied SMR over an $Al_2O_3$ supported Ni catalyst, which was previously modified with $CeO_2$ and $La_2O_3$. They observed that the morphological characteristics (i.e., surface

area, nickel dispersion) are responsible for the enhanced catalytic properties including larger methane conversion and a further decrease of coke formation [31]. As a summary, $CeO_2$, $ZrO_2$, and their mixed oxides are largely used as supports or support dopants, owing to their high oxygen storage capacity, redox properties, and consequently sufficient resistance to coking, leading to superior catalytic performance compared to conventional $Al_2O_3$ or $MgAl_2O_4$ supports [23].

### 2.5. Ni Coordination State

$NiAl_2O_4$ in the reduced and unreduced state, as well as $NiAl_4O_7$ in the reduced state, are active and stable for methane dry reforming due to the presence of 4-fold coordinated oxidized nickel. The limited amount of metallic nickel in these samples minimizes carbon deposition. On the other hand, the presence of metallic nickel is required for methane steam reforming (Figure 3). $Ni_2Al_2O_5$ in the reduced and unreduced states and $NiAl_2O_4$ in the reduced state are found to be active for methane steam reforming due to the presence of sufficiently small nickel nanoparticles that catalyze the reaction without accumulating carbonaceous deposits [32].

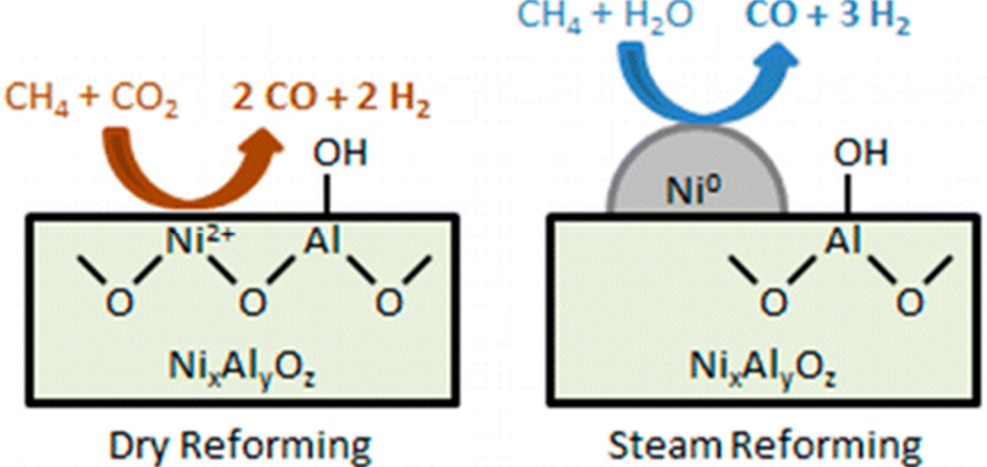

**Figure 3.** Active sites for methane dry reforming and methane steam reforming over nickel aluminate catalysts [32].

### 2.6. Catalyst Acidic/Basic Sites

The typical $Al_2O_3$ support for steam reforming catalysts is acidic and favors cracking and polymerization of hydrocarbons, which is the major reason to accelerate the deactivation of SMR catalysts. Alkali metals such as K and alkaline earth metals such as Mg and Ca are frequently used to improve catalyst stability [33]. Besides using promoters to modify $Al_2O_3$ support, researchers are seeking for alternative supports. Basic supports such as MgO and CaO could also prompt the reaction rate and $H_2$ selectivity via the enhanced sorption effect. Once these supports absorb $CO_2$, the reaction equilibrium is shifted towards to the product side. Based on this effect, an SE–SMR process was developed to pursue higher $H_2$ selectivity [19–21]. Another advantage of the SE–SMR process is the ease of handling absorbed $CO_2$ by decomposing $CaCO_3$ and recovering the catalyst. However, regeneration of CaO is an exothermal reaction.

## 3. Descriptors of Reaction Kinetics

### 3.1. Surface Area of Ni Catalyst

Kinetic parameters are largely influenced by the catalysts and operation conditions. It was assumed that reactant species of $CH_4$ and $CO_2$ were adsorbed onto active sites separately. The adsorbed reactants then associatively react on the active sites and lead to the formation of $H_2$ and CO products. The basic

model is established on the basis that the reactant species of $CH_4$ and $CO_2$ follow the first-order behavior. Therefore, the optimization of reaction conditions and increasing surface area of Ni particles are the key for this aspect.

### 3.2. Binding Energy of Ni–C and Ni–O

The activation of CH4 and the gasification of C species (to CO) are the rate-determined steps for SMR. Especially for the gasification of C species, the stronger bonding of Ni–C will hinder the desorption of C species and generate more carbon deposits (accelerate the deactivation). Previous research work [34] indicates that C or C–H species are the most stable intermediates on the catalyst surfaces. For the reverse reaction (i.e., methanation), the dissociation of CO has a large activation barrier on the perfect Ni(111) surface but is favored on the stepped Ni(211) surface. Hence, the steps on the Ni(111) surface are predicted to be the sites where CO dissociates. The reaction energy, 292 kJ/mol calculated from this DFT study, corresponds to a reaction enthalpy of 230 kJ/mol, which is in good agreement with the experimental value of 206 kJ/mol. The binding energy of Ni–C and Ni–O usually have a volcano-curve, therefore it is important to develop a Ni catalyst with optimized binding energy.

## 4. Descriptors of Reaction Engineering

The steps involved in the SMR process to produce pure $H_2$ can be divided into (a) feed pretreatment, (b) steam reforming, (c) CO shift conversion, and (d) hydrogen purification.

### 4.1. Temperature (Heat Transfer) and Pressure

SMR is severely endothermic ($\Delta H_{298K}$ = +205.9 kJ/mol), and thus thermodynamically preferable under high temperature and low pressure. It has been demonstrated that $\Delta H$ of SMR increases as reaction temperature increases while $\Delta G$ reduces with the increasing temperature. In addition, low reaction temperature leads to coke deposition. Therefore, the SRM process usually runs at a higher temperature (above 700 °C) in order to maintain sufficient reaction activity [1]. For this endothermic reaction, the heat transfer is critical to keep the high reaction rate. Usually, $Ni/Al_2O_3$ powder was further coated on the commercial catalyst support. Hence, material type, shape and coating method all have great influence on the heat conductivity. At same time, the size of the used support is also important for saving the space of the reaction bed. The methane activation is one of the rate determined steps while the coverage of $CH_4$ on the Ni catalyst is crucial. Thus, the reaction should be carried out at a high pressure. On the other hand, due to the increased net number of product molecules, raising the reaction pressure will lower the methane conversion.

### 4.2. Ratio of $H_2O/CH_4$ in Feedstock

The conversion of methane depends on the steam to methane ratio (S/C), which increases with a higher S/C ratio, which varies from 1 to 5. Although the stoichiometry for SMR reactions suggests only 1 mol of $H_2O$ is required for 1 mol of $CH_4$, the reaction in practice is being performed using high a S/C ratio, typically in the range 2.5–3 in order to reduce the risk of carbon deposition on the catalyst surface. At same time, a higher steam ratio will trigger more WGS reaction and change the $H_2$ selectivity. Herein, we listed the several commercial hydrogen production processes from methane in the below table, with the calculated $CO_2/H_2$ ratio. Theoretically, SMR produces one $CO_2$ together with four $H_2$ molecules (5.5 kg $CO_2$/kg$H_2$). For other processes yielding CO as by-product, researchers also consider further converting CO to $CO_2$ using a WGS reaction, and thus calculated the produced $CO_2$ amount, with polyoxometalate (POM) producing the highest amount of $CO_2$ while direct decomposition of methane to solid carbon and hydrogen undergoes a $CO_2$-free pathway (Table 1).

**Table 1.** Summary of different methane conversion processes.

| Reaction | Equation | $\Delta H_{298K}$ (kJ/mol) | Energy Input/mol$_{H2}$ | Reaction Condition | Commercial Catalyst | CO$_2$/H$_2$ (kg/kg) |
|---|---|---|---|---|---|---|
| SMR | $CH_4 + H_2O \leftrightarrow CO + 3H_2$ <br> $CH_4 + 2H_2O \leftrightarrow CO_2 + 4H_2$ | 206 <br> 165 | 68 <br> 41.2 | 700–1000 °C <br> 3–25 bar | Ni/Al$_2$O$_3$ (with promoter) | - <br> 5.5 |
| WGS | $CO + H_2O \leftrightarrow CO_2 + H_2$ | −41 | −41 | HTS [1] <br> 310–450 °C <br><br> LTS [2] <br> 200–250 °C | 74.2% Fe$_2$O$_3$ <br> 10.0% Cr$_2$O$_3$ <br> 0.2% MgO <br> 32–33% CuO <br> 34–53% ZnO <br> 15–33% Al$_2$O$_3$ | - <br><br><br> - |
| DRM | $CH_4 + CO_2 \leftrightarrow 2CO + 2H_2$ | 247 | 123.5 | 800–1000 °C <br> 10–20 bar | Ni or Co-Based | 5.5 |
| POM | $CH_4 + \frac{1}{2}O_2 \leftrightarrow CO + 2H_2$ | −36 | −18 | 400–1000 °C <br> 1 atm <br><br> 1500 °C <br> 125 bar | Ni or Rh-based <br><br> No catalyst | 7.3 |
| Autothermal Reforming | $CH_4 + \frac{1}{3}O_2 +$ <br> $\frac{1}{3}H_2O \leftrightarrow C + \frac{7}{3}H_2$ | 46 | 20 | - | - | 6.6 |
| Methane Decomposition | $CH_4 \leftrightarrow C(s) + 2H_2$ | 75 | 37.5 | - | - | N.A. |
| Hydrogen combustion | $H_2 + \frac{1}{2}O_2 \leftrightarrow H_2O$ | −286 | - | - | - | - |
| Methane combustion | $CH_4 + 2O_2 \leftrightarrow CO_2 + 2H_2O$ | −803 | - | - | - | - |

[1] HTS: High Temperature Shift. [2] LTS: Low Temperature Shift.

We also calculated the energy input (based on enthalpy) for generating one mole of $H_2$. Comparing with the energy output from the combustion of $H_2$ (−286 kJ/mol), all the processes have a net energy output, with POM having the largest amount of released energy, followed by autothermal reforming and then direct decomposition of methane. Therefore, when considering both the amount of released energy and produced $CO_2$, catalytic decomposition of methane to solid carbon is a promising and attractive alternative route for hydrogen production, especially due to its $CO_2$-free feature.

## 5. Catalytic Decomposition of Methane (CDM)

Methane can be directly thermally or thermocatalytically decomposed to carbon and hydrogen without emission $CO_2$ or $CO$, according to the following reaction:

$$CH_4(g) \rightarrow C(s) + 2H_2(g) \Delta H°_r = +74.8 \text{ kJ mol}^{-1}$$

Compared to other traditional hydrogen production from methane based on SMR and methane partial oxidation, which always generates large quantities of $CO_2$, the CDM process has recently attracted the most attention from researchers as it is the most environmentally friendly process for hydrogen [52–55]. For the CDM process, the suitable catalysts play a crucial role which can reduce the activation energy and shorten the reaction time. Typical catalysts includ Ni based, Fe-based, doped noble metals and carbon catalysts [56–59].

Among all the CDM catalysts, Ni-based catalysts were widely investigated due to their high activity [60–62]. It has been reported that hydrogen could already be detected at 200 °C in the presence of a freshly prepared Ni catalyst [63]. Monzón and co-workers reported the results of the properties and catalytic behavior of an $Al_2O_3$ supported Ni catalyst with the loading amount of 30% (30% Ni/$Al_2O_3$) for CDM. The influence of operating and reduction temperatures and the composition of feeding gas (CH$_4$/H$_2$/N$_2$) on methane conversion, hydrogen production and deactivation of catalysts were further investigated. The result showed that the catalyst prepared by co-precipitation performs catalytic activity of methane decomposition reaction above 550 °C. Moreover, the feeding of hydrogen accompanied by methane inhibited both the formation of carbon filaments and coking reaction, which enhances the stability of catalysts to a certain degree [64].

The size of Ni particles plays a critical role in the reaction efficiency of CDM. Takenake et al., studied 40 wt% Ni/$SiO_2$ catalysts with 60–100 nm Ni nanoparticles in CDM, of which the carbon yield was as high as 491 g carbon per gram Ni at 500 °C [65]. As a comparison, Ermakove group prepared a 90 wt% Ni/$SiO_2$ catalyst with Ni particles of 10–40 nm size, which provided 385 $g_C g_{Ni}^{-1}$ at 550 °C [66]. Modification of Ni catalysts with other metals to from bimetallic/trimetallic catalysts is another approach to increase the CDM activity and the stability of Ni catalysts [67,68]. Rezaei, Meshkani and co-workers investigated the catalytic and structural properties of the La, Ce, Co, Fe and Cu-promoted Ni/MgO·$Al_2O_3$ catalysts for thermal decomposition of methane. Compared to other elements, the addition of Cu to Ni/MgO·$Al_2O_3$ dramatically improved catalytic performance with the highest $CH_4$ conversion and $H_2$ yields due to the high activity of the NiCu alloy and fast diffusion of carbon. The results also demonstrated that the Ni–Cu/MgO·$Al_2O_3$ catalyst with 15 wt% Cu showed the highest catalytic activity and stability at higher temperature (>80% $CH_4$ conversion) [69]. Moreover, the methane decomposition activities of rare earth metals (La, Pr, Nd, Gd and Sm) doped Ni–Al catalysts were also explored. The introduction of rare earth elements into Ni/$Al_2O_3$ lead to the formation of a hydrotalcite-like structure which greatly changed the activity of Ni particles. Among them, the Ni/Re/$Al_2O_3$ catalysts present best methane conversion due to the large surface area of Ni particles and the strong interaction between Ni and Re/$Al_2O_3$ [70].

Besides Ni catalysts, Fe-based catalysts are regarded as a promising material for CDM due to their high catalytic efficiency and environmentally friendly features. Furthermore, the partially filled 3d orbitals of Fe can promote the dissociation of hydrocarbon by slightly accepting electrons [71,72]. The $Al_2O_3$ and $SiO_2$ supported Fe catalysts were widely employed in the CDM reaction. Ibrahim and co-workers explored $CH_4$ conversion and $H_2$ yield of CDM at 700 °C over $Fe/Al_2O_3$ catalysts with different Fe loading amounts (14%–63%). The results indicated that at low Fe loading amounts, the yield of $H_2$ increased accompanying the content of Fe increase, which reached maximum of 77.2% with 42 wt% Fe loading amount. However, further increasing the Fe content lowered the hydrogen yield, resulting from reduced catalyst surface area caused by high Fe loading [73]. Similarly, Zhou et al. investigated the catalytic performance of $Fe/Al_2O_3$ with variable loading from 3.5 to 70 for CDM at 750 °C. At the loading amount of 41 wt%, the interaction of $Fe_2O_3$ and $Al_2O_3$ was the strongest, leading to an easy lattice incorporation-like solid solution. Therefore, the 41 wt% $Fe/Al_2O_3$ exhibited the best catalytic activity and stability with 80% methane conversion for 10 h at a reaction temperature of 750 °C. [74] On the other hand, $SiO_2$ is another common support which stabilizes Fe catalysts. Compared to $Fe/Al_2O_3$, $Fe/SiO_2$ always exhibited lower activity but longer lifetime due to the strong interaction between active Fe species and $SiO_2$ supports [75].The Takenaka group compared the catalytic activity of $Fe/Al_2O_3$ and $Fe/SiO_2$ catalysts and found that the carbon yield of $Fe/Al_2O_3$ (22.5 gC $gFe^{-1}$) was higher than that of $Fe/SiO_2$ (7.5 gC $gFe^{-1}$) under the identical reaction condition, which results from the different catalytically active sites ($\alpha$-Fe metal and $Fe_3C$) locally formed during the reaction (Figure 4). For $Fe/Al_2O_3$ with a smaller Fe particle size, $Fe_2O_3$ particles were converted to $Fe_3C$, while $Fe_2O_3$ over $SiO_2$ with a larger size always converted to $\alpha$-Fe metallic species [76].

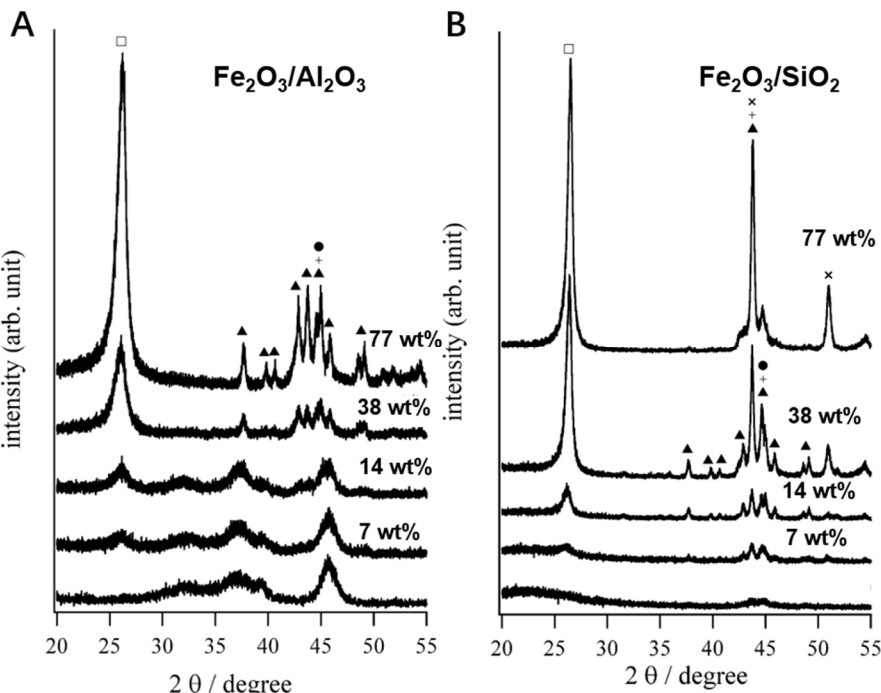

**Figure 4.** X-ray powder diffraction (XRD) patterns of (**A**) $Fe_2O_3/Al_2O_3$ and (**B**) $Fe_2O_3/SiO_2$ catalysts deactivated for catalytic decomposition of methane (CDM) at 1073 K. $\alpha$-Fe metal (●), $Fe_3C$ (▲) $\gamma$-Fe saturated with carbons (×) and graphite (□) [76].

Moreover, supported Fe-based bimetallic catalysts are employed in CDM as well, which always exhibited higher activity and stability than monometallic catalysts [77,78]. Pinilla et al. introduced Mo into Fe/MgO catalysts. The interaction between Mo and Fe particles could inhibit the agglomeration of Fe particles under the operation temperature. Therefore, the Fe/Mo/MgO exhbitied good activity and stability in CDM with 87% methane conversion at 900 °C [79]. The Al-Fatesh group investigated a series of Fe/Ni/MgO catalysts with different ratios of Fe and Ni in CDM, among them the 15Fe/3Ni/MgO displayed the best catalytic property with 73% methane conversion at 700 °C, which was attributed to the presence of a suitable amount of non-interacted NiO species. With the increase in Ni content, CDM activity decreased due to the larger Ni particle size, lower metal dispersion, and thus the lower amount of active sites [80].

With the excellent capability to break up C–H bond, noble metal catalysts were also employed in CDM [81,82]. For instant, Takenaka, Otsuka and co-workers explored the effect of adding different noble metals (Rh, Pd, Ir and Pt) into supported Ni catalysts on the catalytic performance of methane decomposition. Compared to other added elements, the addition of Pd into Ni catalysts significantly improved the catalytic life time and hydrogen yield from CDM, which reached up to 390 $gH_2$ $g(Pd+Ni)^{-1}$ over the catalyst with, a mole ratio Pd/Ni of 1 with the total metal loading of 37 wt% on a carbon nanofiber support. Further experiments explained that the enhancement of activity and stability results from the formation of a Pd–Ni alloy. On the other hand, the pure Ni catalysts will deactivate due to the generation of nickel carbides, while Pd metal particles will fragment into smaller ones during the reaction on Pd catalysts. [83,84] Meanwhile, the Pudukudy group investigated the catalytic performance of Ni/SBA-15 promoted with different Pd loading. The addition of Pd allowed better dispersion of NiO on the support as well as reducing the reduction temperature of NiO due to the hydrogen spillover. Therefore, a maximum hydrogen yield of 59% was observed over the 0.4% Pd catalyst within 30 min, and there was no deactivation observed until 420 min under reaction conditions [85].

Although the CDM has been widely explored with a variety of different catalysts, there are still some challenges that limit its commercial application, especially the rapid deactivation by the produced carbon (coking reaction). In order to avoid the catalyst deactivation, the molten-metal catalyst was proposed by some researchers. Pure molten magnesium (Mg) was applied for $CH_4$ pyrolysis, which achieved 30% of the equilibrium conversion at 700 °C. However, the evaporation of Mg limited its higher conversion at a higher temperature [86]. Metiu, McFarland and co-workers dissolved an active metal (Ni) into an inactive low-melting temperature metal (Bi) to produce stable molten metal alloy catalysts for decomposition of methane to hydrogen and carbon (Figure 5). With the atomic partial negative Ni within the melt, the 27% Ni–73% Bi alloy achieved 95% methane conversion at 1065 °C in a 1.1 m bubble column and produced pure hydrogen without $CO_2$ or other by-products. On the other hand, in the molten alloy system, the insoluble carbon floats to the surface where it can be skimmed off, leading to the high stability of the molten-metal catalyst [87]. Furthermore, they further found that the molten Cu–Bi exhibited better catalytic activity compared to Ni–Bi system, even though neither molten Cu nor molten Bi are good methane pyrolysis catalysts. Further theoretical simulation indicated that the electron-deficient bismuth sites promote the dissociation of methane, leaving the $CH_3$ group bonded to bismuth and H connected to Cu [88].

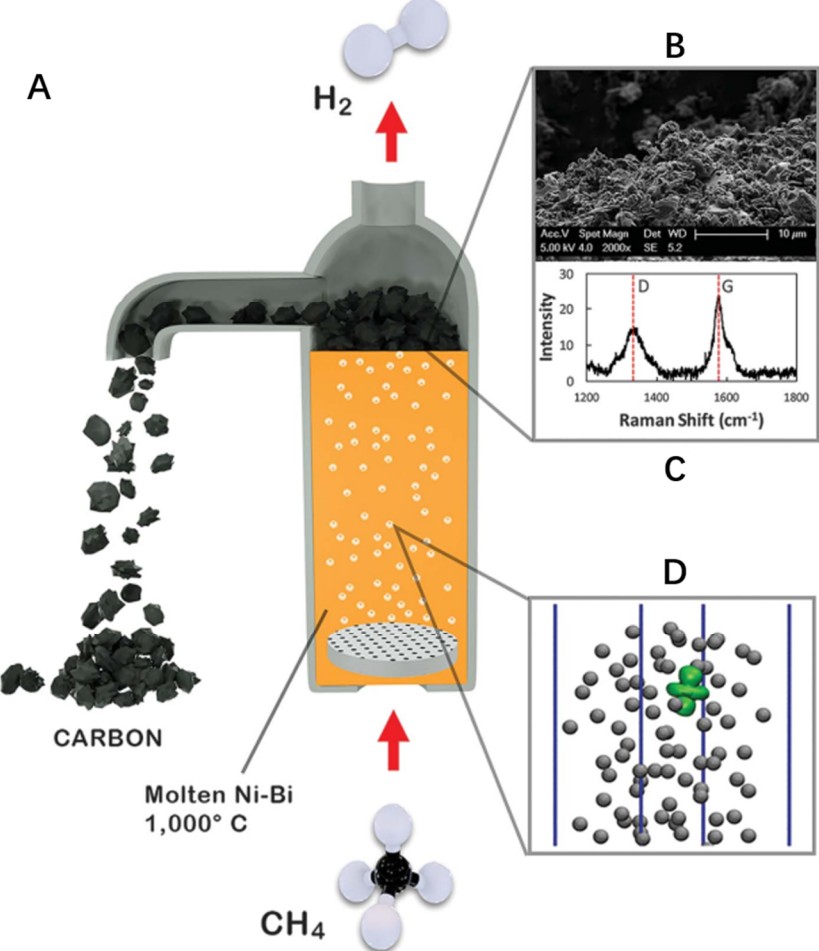

**Figure 5.** (**A**) Reactor for CDM over Ni–Bi molten alloy. (**B**) Scanning electron microscopy images and (**C**) Raman spectrum of produced carbon. (**D**) Ab initio molecular dynamics simulation showing an orbital (green) of a Pt atom dissolved in molten Bi (gray) [87].

## 6. Conversion of Methane to Other Hydrogen Containing Molecules

Compared to methane reforming processes which unavoidably release COx, or CDM process requiring high energy input, converting methane to other hydrogen-containing molecules (e.g., methanol, acetic acid) under a much milder condition followed by hydrogen generation, provides a new approach to utilize hydrogen from methane.

Selective oxidization of methane to methanol, which also contains four hydrogen atoms per molecule, is an efficient method to convert methane to a more active hydrogen-containing molecule. Moreover, as a liquid organic hydrogen carrier (LOHC) [89–91], methanol can release hydrogen by direct decomposition or methanol reforming [92,93]. The Hutchings group prepared a polyvinylpyrrolidone (PVP) stabilized Au–Pd alloy (Au:Pd = 1:1) and investigated its catalytic performance of $CH_4$ selective oxidation in $H_2O_2$ aqueous solution at 50 °C under 30 bar of $CH_4$. Compared to a $TiO_2$ supported Au–Pd alloy catalyst, the colloidal Au–Pd nanoparticles exhibited much better activity, with 92% selectivity at the mild temperature. An isotopical labeling experiment further indicated that most of the oxygenated products were formed from the gas-phase $O_2$ via a radical process, instead of $H_2O_2$ [94]. Xiao and co-workers reported a heterogenous catalytic system with high efficiency for selective oxidation methane to methanol with the in situ generated hydrogen peroxide at 70 °C (Figure 6). They encapsulated Au–Pd alloy particles into zeolite crystals and further modified the external surface of zeolite with organosilanes (AuPd@zeolite-R). The presence of silanes allowed the

diffusion of $H_2$, $O_2$ and $CH_4$ to catalytic active sites while confining the generated $H_2O_2$ to increase the reaction probability. Therefore, AuPd@zeolite-R exhibited excellent performance with 17.3% methane conversion and 92% methanol selectivity [95]. Apart from common nanoparticle catalysts, certain particular methane monooxygenase (pMMO) is an enzyme catalyst for oxidation of methane to methanol in nature [96–98]. For example, extensive studies suggested that the active sites in pMMO are composed of copper complexes coordinated to histidine. Inspired by pMMO, our group employed MOF-808 as a scaffold to host and stabilize highly active copper–oxygen complexes. The catalyst showed high selectivity for methane oxidation to methanol under isothermal condition at 150 °C [99].

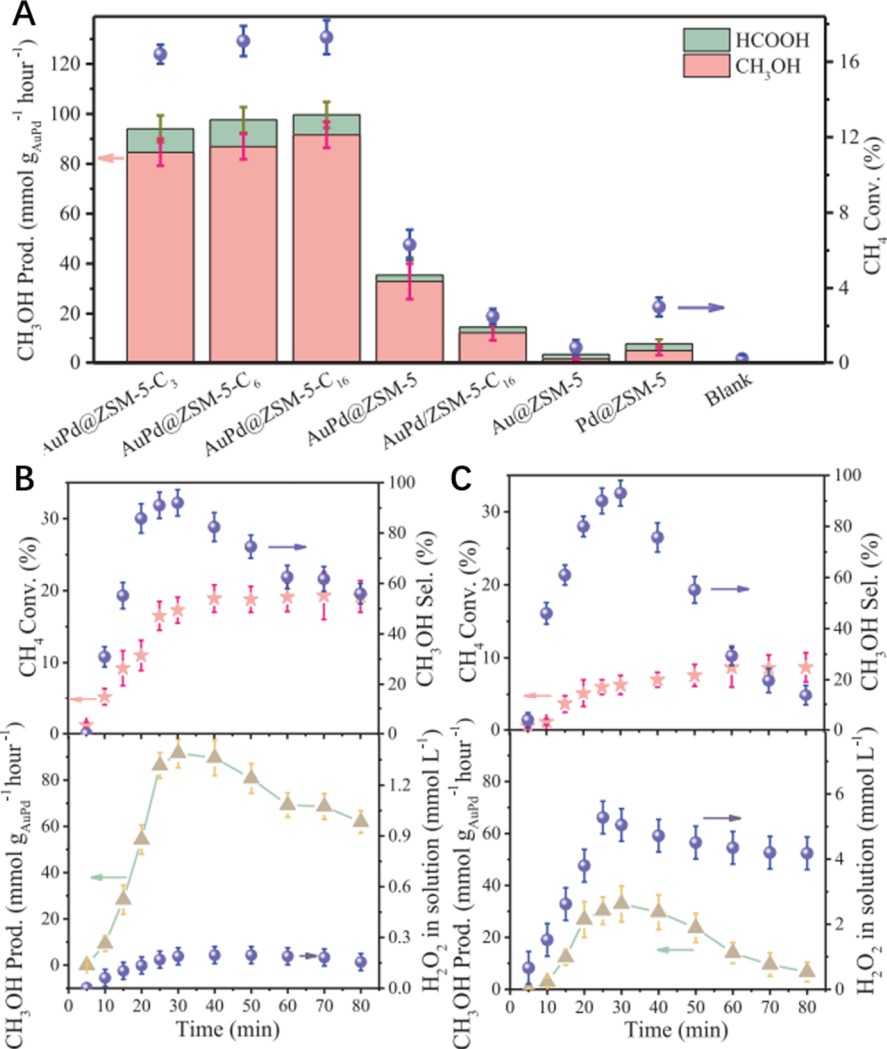

**Figure 6.** (**A**) $CH_4$ conversion and methanol selectivity over different catalysts. Dependences of the methane conversion (Conv.), methanol selectivity (Sel.), methanol productivity (Prod.), and $H_2O_2$ concentration in aqueous solution on reaction time over (**B**) AuPd@ZSM-5-C16 and (**C**) AuPd@ZSM-5 catalysts [95].

Besides methanol, methane could be directly transformed to acetic acid through oxidative carbonylation [100,101], which is regarded as another molecule for efficient hydrogen storage. The Flytzani-Stephanopoulos group employed mononuclear Rh species anchored on zeolite or $TiO_2$ catalysts for direct conversion of methane to methanol and acetic acid in aqueous solution in the presence of $O_2$ and CO at 150 °C. They found that for Rh-ZSM-5 catalysts, the yield of acetic acid was

around 22,000 $\mu$mol/g$_{catalyst}$ with more than 60% selectivity after a 3 h bath mode reaction, which was more efficient than direct conversion methane to methanol [102,103].

## 7. Conclusions

SMR still remains a complicated reaction process, which could be further optimized in terms of catalyst design and operational engineering. Nickel-based catalysts were widely studied due to their high activity and more importantly their low cost. Catalysts with better catalytic performance could be designed by following the descriptors mentioned in this review, including particle size effect, the selection of proper promoters and supports (e.g., $CeO_2$ and $ZrO_2$), and modification of coordination states. Moreover, reaction heat transfer, mass transfer, pressure drop, S/C ratio also need to be optimized for the industrial scale production. Unlike SMR and its derivative processes, which inevitably produce $CO_2$ as by-product, CDM benefits from its higher net energy output and $CO_2$-free reaction pathway. The similar descriptors could be applied in the design of efficient catalysts for CDM as well. Particle size, addition of second metal, supports should all be taken into consideration to optimize the performance of catalysts. Notably, the use of molten metal catalysts provides an effective route for selective $H_2$ production from methane without catalyst deactivation via coke formation, which could be further explored in the future. Furthermore, more research interests were gained to convert methane to methanol as LOHC for easy and long-term hydrogen storage, providing a novel way to utilize methane for hydrogen economy.

**Author Contributions:** Conceptualization, J.S. and G.A.S.; Writing—original draft preparation, L.C., Z.Q., and S.Z.; Writing—review and editing, J.S.; Supervision, G.A.S. All authors have read and agreed to the published version of the manuscript.

**Funding:** The work shown in this paper was supported by Director, Office of Basic Energy Sciences, Division of Chemical Sciences, Geological and Biosciences of the U.S. Department of Energy under contract no. DE-AC02-05CH11231. This work was supported by the Hydrogen Materials Advanced Research Consortium (HyMARC), established as part of the Energy Materials Network by the U.S. Department of Energy, Office of Energy Efficiency and Renewable Energy, Fuel Cell Technologies Office, under Contract Number DE-AC02-05CH11231.

**Conflicts of Interest:** The authors declare no conflicts of interest.

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
