# Peer review of "Catalytic Hydrogen Production from Methane: A Review on Recent Progress and Prospect"

_catalysts, doi:10.3390/catal10080858_

Round 1

Reviewer 1 Report

The paper gives an interesting overview of hydrogen production from methane. It is recommended for publication after minor revisions. Basically, the authors should cite Figures and tables in the text.

Some additional minor changes: line 179 rate-determining steps; line 370: nickel-based catalysts

Author Response

Point 1: The paper gives an interesting overview of hydrogen production from methane. It is recommended for publication after minor revisions. Basically, the authors should cite Figures and tables in the text. 

Response 1: Thanks for the reviewer’s kind reminder. We have cited the Figures and Tables in the text.

Point 2: Some additional minor changes: line 179 rate-determining steps; line 370: nickel-based catalysts.

Response 2: Thanks for the reviewer’s advice. We have made the corresponding changes in the place where the reviewer pointed out.

Reviewer 2 Report

This paper gives a review of the catalytic routes for methane reforming for hydrogen production. The authors introduce the hydrogen production processes via methane reforming and then dig into the catalytic pathways – focusing on Ni-based catalysts – while describing the kinetics and the effective factors regarding the applicability and performance of the catalysts. Finally, give an overview of the catalytic conversion of methane into hydrogen-containing molecules. The paper is written in an organized way and gives valuable information and insight to the reader. It is easy to follow the messages that the authors are trying to give to the reader using easy and clear English. It is a nice work, however, there are a few points to address before consideration for the publication.

Major comments:

  1. It is suggested that the authors address “catalysis” more clearly in the abstract, and if possible, in the title of the paper. The body of the paper and the conclusion part are well written and address clearly the scope of the paper. But the abstract and the title do not well address the scope of the journal. Using more of the keywords like catalysis, catalytic routes, catalytic reforming, etc. may help.
  2. The figures and tables are not referred to in the text. It is recommended to refer to the figures and tables in the text and briefly describe what they are about and how they are related to the discussions.

Minor comments:

  1. The third affiliation is not assigned to any of the authors. Please revise.
  2. In several places, the subscript is needed. For example, lines 33, 52149, 167, 227, 230, 376, 380.
  3. In table 1: what are HTS and LTS? Please describe or mention them in the caption of the table.

Author Response

Major comments:

Point 1: It is suggested that the authors address “catalysis” more clearly in the abstract, and if possible, in the title of the paper. The body of the paper and the conclusion part are well written and address clearly the scope of the paper. But the abstract and the title do not well address the scope of the journal. Using more of the keywords like catalysis, catalytic routes, catalytic reforming, etc. may help.

Response 1: Thanks for the reviewer’s kind suggestion. We have made some modification on the title and abstract to emphasis more on the catalysis.

Point 2: The figures and tables are not referred to in the text. It is recommended to refer to the figures and tables in the text and briefly describe what they are about and how they are related to the discussions.

Response 2: Thanks for the reviewer’s advice. We have cited the Figures and Tables in the corresponding place in the manuscript.

Minor comments:

Point 1: The third affiliation is not assigned to any of the authors. Please revise.

Response 1: Thanks for the reviewer’s comments. We have checked the information of authors and affiliation and the third is assigned to Professor Gabor A. Somorjai.

Point 2: In several places, the subscript is needed. For example, lines 33, 52149, 167, 227, 230, 376, 380.

Response 2: Thanks a lot for the reviewer’s comments. We have corrected these mistakes in corresponding place in the text.

Point 3: In table 1: what are HTS and LTS? Please describe or mention them in the caption of the table.

Response 3: Thanks for the reviewer’s kind reminder. “HTS” is the abbreviation of “High Temperature Shift”, while “LTS” is the abbreviation of “Low Temperature Shift”. And we have made a comment in Table 1.